# An Action Plan to Facilitate the Transfer of Pain Management Competencies Among Nurses

**DOI:** 10.3390/nursrep15120442

**Published:** 2025-12-11

**Authors:** Litaba Efraim Kolobe, Lizeth Roets

**Affiliations:** Department of Health Studies, University of South Africa, P.O. Box 392, Pretoria 0003, South Africa; roetsl@unisa.ac.za

**Keywords:** action plan, competencies, transfer, pain management

## Abstract

In response to persistent gaps in pain management competencies among nurses in Saudi Arabian teaching hospitals and similar healthcare settings globally, this manuscript presents a developed and validated action plan designed to support the effective transfer of pain management competencies into clinical practice. The action plan was developed to address the critical need for structured, practical strategies that enhance nurses’ ability to apply pain management knowledge in diverse interdisciplinary environments. The action plan was validated through a rigorous three-round e-Delphi technique involving 12 expert panel members, achieving a 75% consensus on its content and structure. The final validated plan includes clear action statements, implementation methods, designated responsibilities, and defined timeframes. The core action statements focus on the following: (i) motivating nurses to pursue further study; (ii) equipping nursing teams with appropriate pain management tools; (iii) developing content-specific, practice-oriented short training programs; (iv) tailoring training to accommodate different learning styles; (v) using diverse teaching methods; (vi) creating strategies to encourage participation in training; and (vii) promoting the application of acquired knowledge in clinical settings. Adoption and implementation of this action plan by nursing leadership are anticipated to significantly enhance the transfer of pain management competencies, ultimately improving patient outcomes. The plan is adaptable for use in similar healthcare settings worldwide, offering a replicable model for strengthening nursing practice through targeted competency development.

## 1. Introduction

Training and education are ongoing and dynamic processes that assess an individual’s capacity to integrate acquired knowledge, skills, and attitudes, commonly referred to as competencies, into practice [1]. Transfer of learning may be broadly defined as the carryover of skills, knowledge, or attitudes developed in one learning context to a new learning environment, such as the application of skills, facts, or concepts learned in theory to practice [2,3]. This ability is measured against professional standards and can be improved through training and reflection. The concept recognizes the primary role of the learner in the learning process, and that true learning occurs when the learner understands, makes sense of, and “owns” the knowledge being learned [3].

It is essential for nurses to effectively apply the knowledge they have gained through their education and training in real-world nursing scenarios. Therefore, training hospitals and nursing institutions should provide comprehensive training programs and opportunities that enable nurses to seamlessly transfer their learning into practical application, equipping them to apply their acquired knowledge across diverse situations [1].

Nurses, due to spending more time with patients compared to other healthcare professionals, significantly influence patient outcomes and well-being through the quality of nursing care they provide [4]. Therefore, nurses are obliged to ensure patient safety by assessing and managing their pain effectively and comprehensively [5,6,7]. Addressing pain management comprehensively contributes to improved patient outcomes, such as care experience, recovery, and satisfaction. Due to the global nurse migration, nurses who have completed their education in their home countries can be found working in various hospitals worldwide, including those in Saudi Arabia. This migration creates a diverse mix of ethnic and cultural backgrounds, as well as a range of educational backgrounds among nurses, which affects their proficiency in pain management skills. Although both Saudi nationals and expatriates who work in two hospitals attend pain management training programs, it has been demonstrated that they fail to adhere to pain management guidelines in implementing their knowledge [8,9]. This highlights the rationale for conducting the study, which aimed to develop an action plan to address the critical need for structured, practical strategies that enhance nurses’ ability to apply pain management knowledge across diverse interdisciplinary settings.

The literature indicates problems with the current approach of pain management among nurses in Saudi Arabian hospitals, highlighting significant gaps in their knowledge and improper attitudes toward pain assessment and treatment [9,10,11]. Hammerschmidt and Manser; Al-Sayagh et al., and Samarkandi [9,10,11] studied the knowledge and attitudes of both Saudi nationals and expatriates regarding pain management. Their results revealed that a lack of knowledge is a common factor leading to poor pain management practices among nurses in Saudi Arabia. They also found a positive link between nurses’ pain management skills and their participation in ongoing nursing education. This suggests that regular training and skill development in pain management could improve the quality of nursing care in Saudi Arabian hospitals, and likely in other healthcare settings worldwide.

While research findings demonstrate nurses’ competencies in pain management [9,10], there is a lack of studies in the Saudi Arabian context that examine actions to promote the transfer of pain management competencies among nurses across various settings. Therefore, the aim was to develop and validate an action plan to facilitate the transfer of pain management competencies among Saudi Arabian nurses. The action plan includes measures to address pain management competencies within the targeted nursing group, identify knowledge gaps, and propose strategies to enhance the transfer of these skills. Ultimately, the goal is to improve nursing practices in pain management and increase patient satisfaction with nursing services. The inadequacy in pain management may stem from challenges in effectively transferring pain management competencies among nurses [9,10,11]. To bridge the theory-practice gap in pain management, educators, supervisors, and clinical facilitators may play a major role in teaching nurses about pain management and the appropriate pain assessment scales to use. They need to understand the characteristics of their students, enabling them to apply training information to improve pain management [12,13]. For trainers, understanding trainee characteristics greatly enhances their ability to facilitate the transfer of learning in pain management education. This essential synthesis is supported by research showing that nursing educators are obliged to adhere to the principles of transfer of learning, which include (1) choosing appropriate teaching methods, (2) recognizing students’ characteristics and learning preferences, (3) determining the content of learning, and (4) fostering a conducive learning environment [14,15,16,17].

Despite the high priority given to pain management education in Saudi Arabian teaching hospitals, nurses continue to exhibit inadequate pain management competencies [8]. One potential rationale, among others, could be the challenges associated with transferring pain management knowledge and skills to nurses in clinical settings. Undergraduate programs often do not include pain management as a core subject in their curriculum, or when they do, it receives only limited attention or input [18,19]. When they do, it receives limited attention or emphasis [20]. This could potentially be a contributing factor to why nurses may not prioritize pain management. To address this issue, certain teaching hospitals in Saudi Arabia have established nursing education centers that include pain management specialists, such as pain team members or specialist nurses. These professionals are tasked with teaching other professional nurses [21]. In at least two hospitals in Saudi Arabia, pain team members conduct bi-monthly pain management workshops, each lasting four hours, at the Center for Nursing Education (CNE). These pain management teaching courses are formally conducted at the CNE and in the respective units as unit-based in-service education opportunities [18]. In New York, United States, a similar situation was also experienced, as indicated in a study conducted by Campbell [19] that compared pain management training provided as unit-based education and at centralized education centers. An Ethiopian university hospital introduced a nurse-based pain management program [20], which provided intensive in-service education for nurses to improve nurses’ knowledge and attitudes towards pain.

Insufficient pain management results in patient dissatisfaction, prolonged hospital stays, higher readmission rates, and increased financial expenses [21]. Patients also experience reduced ability to perform physical and complex cognitive tasks due to fatigue, sleep deprivation, anxiety, and depression caused by inadequate pain relief [21,22]. Multiple surveys have provided evidence indicating that patients have expressed dissatisfaction with the management of pain. Consequently, there is considerable emphasis on providing guidance on pain management to nurses to enhance patient outcomes and satisfaction. Several countries, such as Brazil, have initiated comprehensive national programs under the Ministry of Health to offer professional education and support in pain and palliative care [23]. Similarly, in the United Kingdom, the Core Standards for Pain Management Service (CSPMS) place significant importance on professional education in pain treatment [24,25].

The available literature and evidence identified justified the need to develop an action plan to enhance the transfer of learning of nurses’ pain management competencies. The study applied Donovan and Darcy’s Systemic Model of Transfer of Learning (Figure 1) [17] in developing the action plan to facilitate the transfer of pain management competencies among nurses. This was done to improve the effectiveness of pain relief for patients within healthcare institutions.

The rationale for adopting this model was based on the perspectives of Donavan and Darcy [17], who assert that learning is systemic and explores interrelated factors within a specific system that influence the transfer of pain management competencies among nurses in their respective care areas. This model can be employed in postgraduate nursing education to facilitate the integration of pain management theory with specific nursing practice that facilitates nurses’ competencies. The concept ‘competency transfer’ is defined as the application of knowledge, skills, and attitudes acquired in learning contexts to practical settings [17]. In the context of this research, ‘competency transfer’ is operationally defined as the demonstration by all nurses of acquired knowledge, skills, and attitudes relevant to delivering nursing care focused on pain management within their various areas of practice.

The four dimensions of the model were applied to utilize the identified nurses’ characteristics, design pain management education programs, assess the transfer of learning climate within hospital nursing services, and evaluate the working environment [19]. By facilitating the transfer of pain management competencies, it aims to promote best practices and enhance nurses’ professional development, potentially benefiting patient outcomes [15,16,17]. The knowledge and skills acquired from pain management training can be shared with other colleagues, further extending the impact [26]. The literature review and action plan development process followed the framework of the Systemic Model of Transfer of Learning, ensuring alignment and effectiveness in the development of the draft action plan.

The significance of applying this developed action plan extends beyond Saudi Arabian teaching hospitals to include non-Saudi settings as well. Its implementation can enhance nurses’ pain management skills. It offers a reproducible framework for improving pain management in nursing through targeted competency development and the dissemination of best practices. Sharing the action plan could benefit fellow nursing professionals and hospital administrators. Consequently, student nurses, registered nurses in training, and clinical facilitators should utilize both effective and ineffective transfers to bridge the gap between pain management theory and real-world practice. Additionally, supervisors, preceptors, patients, family members, and others involved in daily pain management nursing care play a crucial role in fostering patient-centered care, promoting dignity, independence, and choice in nursing practice. The review of teaching and assessment methods also aims to facilitate the transfer of pain management competencies among nurses. An action plan is a written or systematic approach that outlines specific steps to achieve set goals, identifying all key stakeholders responsible for executing these activities [27]. The goal of the action plan was to develop a comprehensive strategy to facilitate the transfer of pain management competencies for nurses working in Saudi Arabian teaching hospitals. As previously mentioned, the development of the action plan was motivated by the recognition of existing competency gaps within these two teaching hospitals.

Within the study’s scope, an action plan was developed for implementation by the nursing administration at two Saudi Arabian teaching hospitals. The plan acts as a vital tool to help transfer pain management skills among nurses. Based on data obtained from nurses and other relevant stakeholders [28], the plan offers structured, detailed guidance for implementation, monitoring, and continuous improvement. It outlines steps for clinical facilitators and nurses to follow to reach specific goals, such as enhancing pain management skills. This includes pinpointing needed resources and establishing a timeline for transferring these skills [29]. The action plan details the action statements, methods, responsible parties, and deadlines for each step. This may lead to greater patient satisfaction, better nursing care experiences, faster recovery, and reduced complications related to uncontrolled pain control through adopting this plan [21].

## 2. Materials and Methods

### 2.1. Aim and Objectives

The study aimed to develop an action plan to facilitate the transfer of pain management competencies among nurses in various interdisciplinary settings within Saudi Arabian teaching hospitals.

The following objectives were achieved to attain its aim. Firstly, describing accessible resources for conducting pain assessments; secondly, describing nurses’ characteristics and learning styles that enhance the transfer of pain management competencies; thirdly, exploring the teaching methods employed by clinical facilitators in educating nurses on pain management; fourthly, describing the learning content about pain assessment and management; fifthly, delineating the transfer of learning climate within hospital nursing care areas [28], and finally, developing an action plan to facilitate the transfer of nurses’ learning regarding pain management competencies.

### 2.2. Design

A mixed-methods approach, following an explanatory sequential design as outlined by Creswell [30], was used in this study to gather evidence for the development of an action plan aimed at facilitating the transfer of learning in pain management skills among nurses. The current manuscript focuses specifically on the qualitative study, which involved the development and validation of the action plan to ensure that the voices of all relevant stakeholders were considered. Figure 2 presents an overview of the processes that informed the development of the action plan.

The development of the action plan to facilitate the transfer of pain management competencies among nurses was guided by several key sources: (1) available resources for pain assessment (quantitative phase); (2) nurses’ characteristics and learning styles (quantitative phase; (3) teaching methods, learning content, and learning climate within hospitals (quantitative phase; (4) a comprehensive review of the literature; and (5) all of these elements were connected to the resulting action plan, which was then validated using the E-Delphi technique (qualitive phase). Figure 3 illustrates the step-by-step process of developing the action plan, highlighting how the various sources of evidence were integrated into the process.

### 2.3. Setting

The study sites, where the data that informed the development of the action plan, were two (Hospital A and B) purposively selected teaching hospitals in Riyadh, the Kingdom of Saudi Arabia. Five nursing care divisions, which were purposively chosen based on their fulfilment of the highest eligibility criteria, were involved. These nursing care divisions represented were 19 (nineteen) medical wards, 9 (nine) surgical wards, 9 (nine) paediatric wards, 4 (four) cardiac wards, and 7 (seven) obs-gynae (obstetric-gynaecologist) wards. In Hospital A, it was fifteen medical wards, eight surgical wards, four cardiac wards, one pediatric cardiac ward, and five obstetrics-gynaecology wards. In Hospital B, four medical wards, one surgical ward, and eight paediatric wards were represented.

### 2.4. Population and Sampling

To provide a clear understanding of the stages leading up to the development of the action plan, it is essential to reference the population and sample sizes associated with the earlier phases used to collect evidence, as illustrated in [28].

To support the necessary evidence for developing the action plan, a population of 385 registered nurses (RNs) and 47 clinical facilitators (CFs) was relevant. A proportionate stratified probability sampling method was used to recruit RNs and CFs, aiming to adequately represent the five nursing care divisions within the two teaching hospitals. The Raosoft 2011 sample size calculator was employed to determine the size of each stratum and the overall sample size, ensuring that a margin of error and confidence level were calculated [29,31,32,33]. This justifies the proportionally allocated sample size drawn from RNs and CFs working in the five nursing care divisions of the two hospitals, addressing selection bias. Among the population of 1462 registered nurses and 47 clinical facilitators across the two hospitals, wards were classified into five nursing care specialties to create strata. These five strata included medical wards, surgical wards, pediatric wards, cardiac wards, and obstetrics-gynecology units. Based on the 1509 registered nurses and clinical facilitators in all five strata, the sample size was then calculated for each specialty using the Rao-Soft sample calculator to achieve a 95% confidence level, as illustrated in Table 1.

This method guarantees that all qualified RNs and CFs have an equal chance of being selected, thus reducing potential bias.

The inclusion criteria for participants were as follows:Attendance at a pain management workshop within the last three years.Receipt of in-service pain management training within the past year.Training in pain management by clinical facilitators for nurses in the five nursing care divisions.Comfort with being interviewed in English.

In total, 12 e-Delphi panelists, 10 registered nurses (RNs), and 2 clinical facilitators (CFs) were purposively selected for inclusion based on their expertise.

The criteria for including panel members were as follows:Demonstrated enthusiasm for pain management.Registered nurses who have had at least one pain management training course within the past two years.Registered nurses who received pain management in-service training within the last year.Clinical facilitators who are responsible for instructing nurses in pain management within the nursing care divisions.Willing to participate in a minimum of three Delphi rounds.

### 2.5. Ethical Considerations and Data Collection

The research was ethically approved by the Research Ethics Committee of the custodian University (Ethics certificate number REC-012714-039) and the King Abdullah International Medical Research Committee (KAIMRC) (SP 18/036/R). Those panelists who met the inclusion criteria participated voluntarily by accessing a secure link provided in a recruitment letter, which directed them to the action plan and accompanying validation instrument. The software platform used ensured anonymity by providing only aggregated, anonymized data to the researchers.

### 2.6. Data Collection

The development of the action plan was guided by a six-step framework adapted from McCurk and Mueser (2021) [34]. Since this study employed the Delphi technique, the report adheres to the CREDES (Conducting and Reporting DELphi Studies) guidelines, as recommended by Niederberger and Renn (2023) [35].

Justification of the method: The process began by identifying challenges related to the transfer of competency, based on the lowest-rated items reported by nurses and facilitators, as previously described. All data were analyzed in relation to the key components of the Systemic Model of Transfer of Learning by Donavan and D’Arcy, and further supported by an in-depth literature review. This process resulted in the formulation of seven action statements, which are presented in Table 2.

A validation tool was developed to assess each action statement and item within the draft action plan. Implemented via Google Forms, the tool included closed-ended (tick box) options to indicate agreement or disagreement with each statement, along with open-text fields for comments and responses to open-ended questions. This format enabled panel members to provide detailed written feedback on the draft action plan.

### 2.7. Validity and Reliability Pre-Testing of the Delphi Validation Tool

Before data collection for the actual Delphi techniques commenced, the validation tool underwent pre-testing and necessary adjustments [36,37] to assess its content and face validity before being distributed to the main study panelists. The items were meticulously crafted, considering the comprehensive literature review pertinent to the draft action plan.

Panel selection during pre-testing: Two registered nurses and two clinical facilitators from non-sampled nursing divisions were invited to participate in pre-testing the tool.

Feedback mechanism during pre-testing: Upon receiving the contributions of all four panelists, the feedback was incorporated, and a comparable draft was provided to them again for the retest. In both rounds, a 100% response rate was achieved. The Delphi validation tool underwent content and face validity assessments. At the same time, reliability was established through three rounds of e-Delphi conducted via Google Forms, achieving a 75% consensus, a similar process to that in the main study.

The panel selection of the main study: The panel for the Delphi in Phase 5 consisted of ten registered nurses and two clinical facilitators, all selected based on their expertise and knowledge in pain management.

Round-by-Round Structure: Three rounds were conducted through independent and anonymous inputs in each round, via email and a link accessible from recruiters’ letters, with nurse managers serving as gatekeepers and judging them to be knowledgeable about pain management.

Anonymity: This ensured that panelists were anonymous while providing their inputs and suggestions in Google Forms.

Feedback mechanism: The aggregation of the group’s inputs was objectively assessed and analyzed in each round. Disagreements or suggestions in each round were noted and implemented when an item did not achieve 75% consensus among all panelists.

Consensus criteria: Items that failed to achieve consensus were selected for the subsequent round until a 75% agreement was attained.

Iterative nature: Panelists, during the second and third rounds, were requested to respond only to those items on which an agreement had not been reached during the previous round. This is evident in the results of each round in Table 2, which indicate those items that scored the highest in terms of views. Panelists, during the second and third rounds, were requested to respond only to those items on which agreement was not reached during the previous round. This is evident in the results of each round in Table 3, which indicate the items with the highest view scores.

Transparency: After each round, the panelists’ aggregated responses were objectively analyzed. Any items that did not receive at least 75% agreement from the panelists were revised based on their suggestions. These revised items were then included in the next round of the Delphi process. This process continued until every item reached a minimum of 75% consensus.

### 2.8. Trustworthiness

Rigor was assessed by trustworthiness. The principles of credibility, confirmability, transferability, dependability, and authenticity, as described by Polit and Beck, and the adoption of the CREDES reporting technique ensure the quality and rigor of the Delphi technique [35,36].

### 2.9. Data Analysis

The e-Delphi data analysis encompassed qualitative and quantitative data. The data were subjected to descriptive statistical analysis as provided by the Google Forms software program of Microsoft Foms 365. The results were presented in the form of frequencies and percentages. Additionally, the comments and suggestions provided by the panelists were subjected to open coding through thematic content analysis, following the methodology proposed by Clarke and Braun [38]. After each round of Delphi, within the validation tool, all items where consensus had been reached were marked “consensus reached.” Recruitment letters with links were sent to gatekeepers for distribution to panelists to obtain access. Similar to the previous round, the validation instrument for the next round was again loaded onto Google Forms. On the validation instrument, these steps were repeated until a 75% consensus was reached. The input from the panelists in the three Delphi validation rounds is thoroughly presented through quotations and an analysis of divergent perspectives in Table 2. 

## 3. Results

### 3.1. Biographic Data

Twelve-panel members voluntarily validated the draft action plan, consisting of ten registered nurses (83.3%; n = 10) and two clinical facilitators (16.7%; n = 2), with ages spanning from 26 to 47 years old. Table 3 offers an overview of the additional demographic characteristics of the participants. These demographics serve to familiarize the reader with the context.

The findings of this report encapsulate the perspectives and input of the 12 panelists listed in Table 3 about issues that achieved the most significant consensus over the three Delphi rounds, culminating in a 75% agreement. The results of the panelists’ opinions in Table 3 are presented just with the issues that achieved the greatest percentage of agreement, as well as those that failed to reach consensus for further rounds. In rounds 1 and 2, all panelists (N = 12) validated the tool; however, in round 3, only 83.3% (n = 10; N = 12) of panelists completed the round. 

### 3.2. The Final Validated Action Plan

After adhering to the six processes of action plan development [34] and integrating the literature review as well as the findings from the preceding three phases (1 to 3), the three rounds during Delphi validation final agreed on the action plan illustrated in Table 4. The items that the panelists reached a consensus of 75% and more are articulated in the final action plan, see Table 4, and these include the seven action statements that encompass the following: (1) motivating nurses to pursue further studies; (2) providing the nursing team with appropriate pain management tools; (3) developing a practice-oriented, content-specific short training program for pain management; (4) developing a short pain management program that accommodates all learning types; (5) incorporating different teaching approaches to accommodate diverse learners and facilitators in the training of pain management; (6) developing strategies to motivate nurses to participate in the short training program; and (7) motivating nurses to apply the knowledge gained in the training program into practice. These items include the methods suggested to achieve the action statement and expected outcomes, the responsible persons to enhance the actions suggested, and the frame to achieve the suggested methods.

## 4. Discussion

As an ongoing process, action planning was considered an important tool for determining success based on ideas, the resources required to accomplish targets, and deadlines for completing tasks assigned to responsible individuals or organizations [39]. As can be seen in Table 3 and Table 4, the action plan was developed following a thorough consultation process of Delphi validation in phase 5, which included participation of 10 registered nurses and 2 clinical facilitators who were considered to have expert opinions. Therefore, implementing this validated action plan is essential in other settings to advance practice beyond current interventions.

### 4.1. Motivating Nurses to Further Their Studies

Several factors can serve as motivation for nurses to pursue further education to enhance their educational competencies [40]. The panel members concurred that for extrinsic motivation of nurses would entail implementing a certificate system to acknowledge distance learners, granting a day off to attend a pain management program, providing a financial reward upon completion of a pain management course (degree or diploma), and offering complimentary lodging during study leave, presenting and championing the adoption of the policy to the Ministry of National Guard Health Affairs (MNGHA) via the Central Region Nursing Governance and Accountability Board, as well as integrating the policy into the protocols of all hospitals. The results are consistent with similar research findings, highlighting those individual attributes, such as holding a bachelor’s degree, hold greater significance than a diploma in enhancing nurses’ knowledge. These attributes also act as incentives for nurses to enhance their knowledge and pursue further studies [39,40,41]. Motivating nurses to pursue studies beyond a bachelor’s degree may improve their nursing practice through experiential learning and the acquisition of new knowledge.

It appears that nurses need intrinsic motivation to pursue higher education, advance their studies, and undergo training to improve their expertise in pain management [40,41,42,43]. Such motivations can include badges, certifications, rewards, and degrees, serving as resources to encourage adult learners involved in distance education [44,45,46]. Being intrinsically and extrinsically motivated is described by scholars specifically referencing the interest in distance learning institutions such as the University of South Africa, the University of the United Kingdom, and the German Fern University to be used for furthering learning at a distance [47].

### 4.2. Pain Management Tools to Be Accessible to the Nursing Team in Every Clinical Area

Participants confirmed using systemic pain assessment guide methods, self-reporting, non-self-reporting, and dementia/cognitive impairment patients. Panelists confirmed that tools such as Provocation and palliation symptoms, Quality of pain, Region and radiation of pain, Severity of pain, and Timing (PQRST), as well as Crying, Required oxygen, Increased vital signs, Expression, and Sleeplessness (CRIES), were necessary for pain management. Swan and Hamilton [47] confirm the findings of this study by stating that the International Association for the Study of Pain (IASP) 2021 Article 3 grants all people with pain access to appropriate assessment and treatment by appropriately trained healthcare professionals using pain rating tools such as PQRST and CRIES. As advised by Simonson et al. [45], the action plan included pain management training for nurse supervisors to assist the nursing team in pain assessments in all nursing care areas. Studies [47,48,49,50] confirmed the panelists’ opinion that hospitals must make all internet information available to patients and their families. Nursing teams can deliver comprehensive pain management treatments and access additional support for pain management. Kahsay and Pitkäjärvi [42] stressed that accurate pain assessment and effective pain management depend on the availability of appropriate pain assessment tools. The accessibility of these pain management tools in the clinical areas may enhance nurses’ practice compared to those areas without such resources, as precise pain assessment helps to optimise pain intervention.

### 4.3. Developing a Short, Practice-Based Pain Management Training Program

As suggested by Hadsell et al. [51], effective pain management strategies should be tailored to the individual needs of patients in their respective work environments. All panelists agreed that this needs analysis should be included in the action plan. This analysis focuses on providing pain management training content that focuses on practical application. This action plan provides efficient pain assessment and pain intervention strategies, as well as comprehensive pain assessment in nursing care, utilizing appropriate pain rating tools. The study by Lovasi et al. [52], conducted in Hungarian hospitals, supports the recommendations of this action plan. Additionally, they suggested that nurses could be trained in-house to enhance pain management practice in their study. A short, practice-oriented pain management training program significantly enhances nurses’ competencies, advancing their nursing practice, and optimizes pain interventions.

### 4.4. Developing a Short Pain Management Program for All Learners

As stated by Lengyel [53], developing a customized training plan begins with understanding the trainee’s learning style. This allows the trainer to select the most suitable methodology. Incorporating multiple learning styles into a pain management program helps with the practical application of knowledge. The panel members identified three types of learners for the action plan: creative thinkers, enthusiastic learners, and organized learners. A study by Geleta et al. [54] in Ethiopia confirmed the findings of the current study by showing that trainees learn better when various learning styles are used during training. Additionally, Decampo [54] found that trainers should adapt their teaching strategies based on trainees’ learning styles. The study notes that creative, enthusiastic, and organized thinkers enhance the transfer of learning [53]. Results indicate that pain management programs should accommodate individuals with different learning styles. Tailoring educational approaches to individual learning styles may improve understanding and application, potentially advancing pain management practices among nurses.

### 4.5. Incorporating Different Teaching Approaches in Pain Management Training

Different teaching strategies are necessary to support the application of pain management knowledge learned during training at work [55]. The action plan highlighted the use of focus groups and role-playing as effective teaching methods. A study by Deocampo [55] recommended similar approaches to address the significant knowledge gap and enhance the transfer of learning for healthcare professionals in pain management skills. To improve this transfer, various methods can be employed to suit different learners and facilitators [54]. Consequently, sharing knowledge about pain management can empower nurses to become more flexible and resourceful, boosting their critical thinking and enabling them to better apply their expertise in pain interventions.

### 4.6. Developing Strategies to Motivate Nurses to Attend Short Training Programs

Findings show that the panel agreed on ways to motivate nurses to participate in short pain management programs. These findings align with He et al. [56], who emphasize the importance of infrastructure and organizational support, such as refresher training and flexible working arrangements. In addition, it highlighted the need to engage nurses in home health care and training programs, while also recognizing the economic factors and sharing the benefits with nurses. This action plan was developed based on the panelists’ agreement that nurses should contribute to developing content, goals, and training by communicating the advantages of pain management competencies and by creating a supportive environment. Based on the study [56], applying self-regulated learning theory (SLR) involves nurses in designing the training content and goals to promote deep and experiential learning. Participants in the study indicate that intrinsic motivating factors, such as self-efficacy, self-directedness, self-awareness, and metacognitive skills, improve self-regulated behaviors. It is evident from the findings of this action plan that transferring knowledge and skills in pain management is essential. A previous study [57] also emphasized the importance of highlighting these competencies. The study demonstrated that the participants acquired the necessary foundational skills for clinical practice [58].

### 4.7. Motivating Nurses to Apply Their Training to Practice

Finally, the action plan recommends implementing strategies agreed upon by the panel members to increase nurses’ intrinsic motivation to apply the knowledge they gained from the pain management training program. Offering pain management experts the chance to take on expert roles, supporting specific, measurable, attainable, realistic, and time-bound (SMART) goals, identifying what motivates nurses to use their knowledge, and awarding grades during annual evaluations are all examples of approaches that both intrinsically and extrinsically motivate nurses to transfer pain management competencies. Kivinity et al. [59] conducted their study in three primary and specialized medical care organizations in Finland and suggested that competence requires practicing roles and responsibilities at the beginning of training. Li et al. [60], in a study in Japan, found that SMART goal orientation best facilitates trainees’ self-directed learning. Self-directed learning influences trainees’ willingness to apply the knowledge they gain. An Ethiopian study by Ayalew et al. [61] highlights the urgent need for policymakers to reward professional nurses based on their performance ratings to improve job satisfaction and motivation. This emphasizes the extrinsic motivation for nurses to apply what they have learned about pain management in their workplaces. The findings of these studies support the recommendations in the action plan that nurses should be motivated both intrinsically and extrinsically to participate in short pain management programs and to assume roles in pain management. Applying training to practice enhances nurses’ pain management competencies and improves nursing practice compared to those without training.

### 4.8. Implications

Nursing care areas will benefit from adopting and implementing the action plan developed from this study’s findings, which will facilitate nurses’ ability to transfer their knowledge of pain management and facilitate their competence. Continued research is recommended to evaluate the impact of training and development strategies on nurse competencies in pain management.

### 4.9. Limitations

This manuscript has limitations that warrant acknowledgment. It focuses primarily on the developed action plan and the methods and results of its validation. A detailed account of the original findings in [28], is not included due to space constraints and the extensive nature of the original data.

Additionally, while there are numerous teaching hospitals in the Kingdom of Saudi Arabia, the study was limited to only two, which were purposefully selected. This presents a methodological limitation, as the findings may not fully represent the broader healthcare context.

The action plan developed in this study is not intended for direct generalization. However, it may be adopted or adapted in similar settings to enhance nurses’ competencies in pain management.

## 5. Conclusions

An action plan was developed with the intent of facilitating the transfer of learning and abilities related to pain management among nurses in Saudi Arabian teaching hospitals. Key components of the action plan for facilitating competency transfer in pain management include fostering nurses’ motivation to pursue ongoing education and participate in targeted short training programs, as well as developing practice-oriented and role-specific content tailored to their clinical areas. Securing support from nursing leadership is critical to ensuring the successful implementation of the initiatives.

Given that the action plan is grounded in evidence from the literature and expert consensus obtained through Delphi rounds, its applicability may extend beyond Saudi Arabia, potentially informing nursing practice in other contexts.

The implementation, adoption, or adaptation of the action plan has the potential to improve nurses’ competency in pain management transfer, provided there is sufficient institutional readiness and support for its success.

## Figures and Tables

**Figure 1 nursrep-15-00442-f001:**
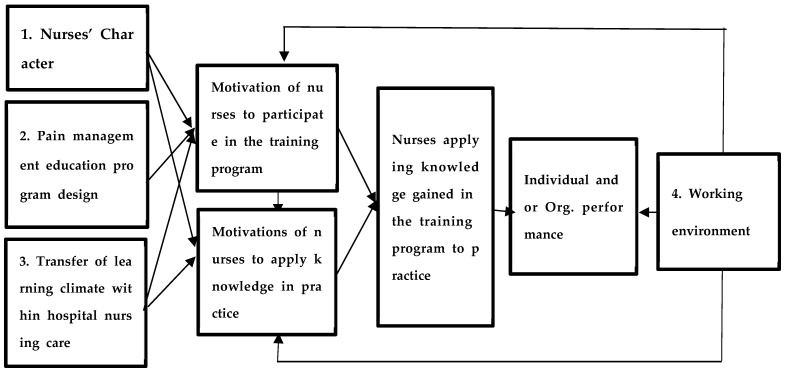
Application of the Systemic Model of Transfer of Learning, as aligned from the citation by Donovan and Darcy (2011, p. 125) [15].

**Figure 2 nursrep-15-00442-f002:**
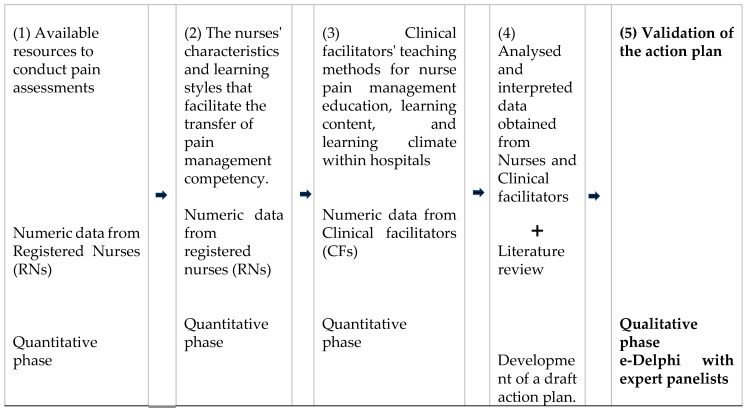
Illustrative diagram of the process of developing an action plan applying the Systemic Model of Transfer of Learning.

**Figure 3 nursrep-15-00442-f003:**
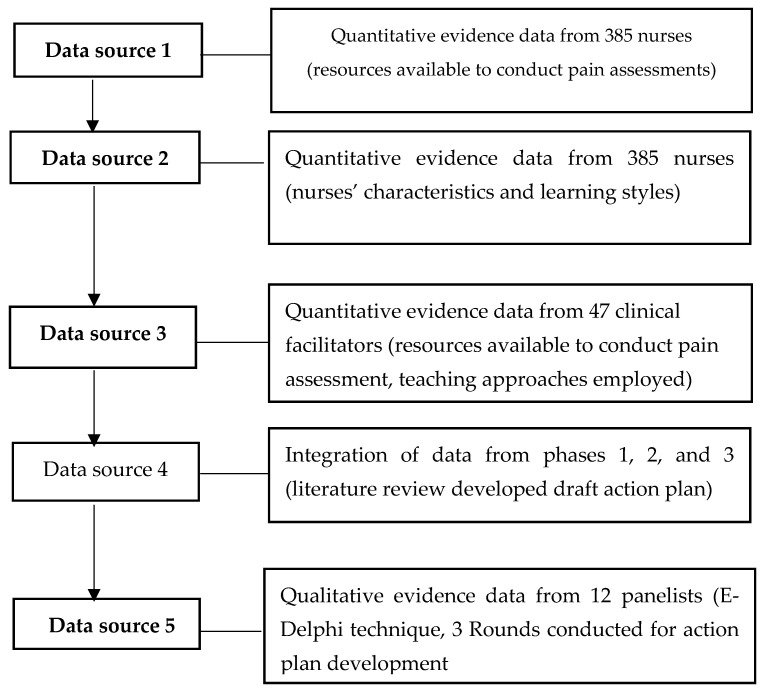
Illustrate a flow chart showing the mixed method and participant flow across phases.

**Table 1 nursrep-15-00442-t001:** Sample size calculation of registered nurses and clinical facilitators in five nursing strata of the two hospitals.

Wards	Medical		Surgical		Paediatric		Cardiac		Obs-Gynae		Total	
**Hospitals**	A	B	A	B	A	B	A	B	A	B	A	B
**Population RNs**	459	135	282	35	20	216	166	0	114	35	1041	421
**Sample size RNs**	210	101	163	33	20	139	117	0	89	33	281	202
**Population of CFs**	14	5	8	1	8	8	4	0	5	1	32	15
**Sample size of CFs**	14	5	8	1	8	8	4	0	5	1	30	15

**Table 2 nursrep-15-00442-t002:** Illustrative Delphi rounds results indicating the responses of the panelists.

		Round 1 (N = 12; f = 100%) Consensus reached Yes/No = (≥75%)	Round 2 (N = 12; f = 100%) Consensus reached Yes/No = (≥75%)	Round 3 (N = 10; f = 100%) Consensus reached Yes/No = (≥75%)
**1**	**1.1 Action Statement:** **Motivate nurses to pursue further studies**	100% consensus reached that nurses must be motivated to further their studies (Yes)	Yes	Yes
	**1.2 Methods**	100% consensus that a policy must be developed to motivate nurses to improve their nursing qualifications. (Yes)	Yes	Yes
		Items to be included in the policy to motivate nurses were: 100% of respondents agreed that a certificate should be issued as an acknowledgement 83.3% agreed that one day off to attend a one-day pain management program 75% indicated that a monetary incentive after completion of the pain management program	Yes	Yes
		91.7% (n = 11) consensus reached that the policy should be presented and negotiated for the implementation to motivate nurses in their nursing qualifications (Yes)	Yes	Yes
		100% consensus was reached to include the policy as part of the hospitals’ polices after approval. (Yes)	Yes	Yes
	**1.3 Responsible persons**	83.3% (n = 10; N = 12) agreed that the clinical director of nursing operations and the director of nursing education should be persons to develop the policy to motivate nurses to improve their qualifications. (No)	Only 41.7% (n = 5) of panelists indicated that a nursing policy committee representative should be responsible. (No)	100% (N = 10) of panelists agreed that the nursing policy committee representative should be responsible for developing the policy that motivates nurses to further their nursing qualifications
		75% (n = 9; N = 12) agreed that the nursing policy committee representatives. (Yes)	Yes	Yes
		83.3% (n = 10) agreed that clinical directors of nursing operations and nurse managers in all nursing care areas to be responsible for including the policy in all hospitals’ policies. (No)	66.7% panelists indicated nurse managers in all nursing care areas to include the policy in all hospitals’ polices to motivate nurses to improve their nursing qualifications (No)	90% of panel members agreed that Clinical directors of nursing operations in every facility are responsible for including the policy in all hospitals’ policies
	**1.4 Time frame**	75% (n = 9; N = 12) agreed that 4–6 months was needed for policy development. (Yes)	Yes	Yes
		No consensus reached by panelists on the time frame to present the policy and for its implementation (No)	66.7% of the panelists indicated the time frame to present the policy and for its implementation within 6 months (No)	90% consensus was reached that 6 months is a required time frame to present the policy and negotiate for the implementation (Yes)
		No consensus reached as panelists had diverse opinions on the time frame to include the policy in all hospitals’ policies after approval by the hospitals (No)	75% (n = 9) of panelists reached consensus to include the policy within 3 months after approval of the action plan (Yes)	Yes
	**Quotes**	*“When there’s a policy, the organization will encourage nurses by giving them a day off or an education day.”* *“A policy must specify that an employee should have worked for the organization for at least two years.” P1* *“A policy must specify that an employee should have worked for the organization for at least two years before he or she can apply for this course.” P2*	None	None
**2**	**2.1 Action statement: Provide the nursing team with appropriate pain management tools**	91.7% consensus reached that an appropriate and relevant pain management tool must be made available (Yes)	Yes	Yes
	**2.2 Methods**	No consensus was reached as the panelists had diverse views on including pain assessment tools in an electronic patient record system (No)	Yes	90% (n = 9) of panelists agreed that the pain assessment tools should be included in the electronic patient record system (Yes)
		Tools to be accessible on the electronic were: 66.7% selected the PQRST guide assessment tool 66.7% panelists chose the CRIES pain scale (No)		
		83.3% consensus was reached to involve nurse supervisors with pain management training and supervisory support. (Yes)	Yes	Yes
		91.7% of paternalists reached consensus that internet-based resources should be accessible to the patients and family members (Yes)	Yes	Yes
		91.7% consensus was reached that the hospital’s intern-based resources on pain management publications should be accessible to nursing staff (Yes)	Yes	Yes
		Internet-based resources that should be included were: 83.3% of panelists agreed that patient pain management websites and support groups should be accessible (Yes)	Yes	Yes
		100% consensus was reached that hospitals’ internet-based resources on pain management publications and electronic materials should be accessible to nursing staff (Yes)	Yes	Yes
		Internet-based resources to be accessible were: 83.3% selected pain toolkits 75% agreed to be videos on pain management and clinical updates (Yes)	Yes	Yes
	**2.3 Responsible persons**	91.7% panelists indicated that the associate director of informatics to include pain assessment tools in the electronic patient record system 75% agreed on one pain nurse specialist to include pain assessment tools in the electronic patient record system (No)	75% panelists agreed that one pain nurse specialist should be appointed in every facility to ensure that the electronic format on pain assessment tools is included in the electronic patient record system (Yes)	Yes
		75% indicated the clinical directors of nursing operations and clinical facilitators in all nursing care areas as responsible persons to involve nurse supervisors with pain management training to provide supervisory support to the nursing staff (No)	50% (n = 6) of the panelists selected the persons to be clinical directors of nursing operations and clinical facilitators in all nursing care areas, to involve nurse supervisors to provide supervisory support to nursing staff. (No)	90% indicated that clinical directors of operations to involve nurse supervisors to provide supervisory support to nursing staff (Yes)
		No consensus reached as the panelists indicated diverse views about persons to ensure internet-based resources on pain management should be accessible to the nursing staff in all nursing care areas (No)	58.3% indicated one pain nurse specialist in every facility to ensure internet-based resources on pain management (No)	90% agreed that clinical directors of nursing operations are responsible for ensuring internet-based resources on pain management publications are accessible to nursing staff in all areas (Yes)
	**2.4 Time frame**	58.3% indicated a 4–6-month timeframe to include the pain assessment tools in the electronic patient record system (No)	58.3% of panelists indicated the time frame of 1 to 3 months within which pain assessment tools should be available in the electronic patient record system. (No)	90% consensus was reached that 1 to 3 months should be the time frame within which pain assessment tools should be available (Yes)
		66.7% selected that every patient round should be done when needed as a time frame, to involve a nurse supervisor (No)	No consensus reached on time frame as panelists gave diverse opinions	100% agreed that every shift, when need arises, as a time frame that nurse supervisors should be involved in providing pain management training and supervisory support. (Yes)
		75% panelists agreed that 24 h access 7 days a week, a time frame to make the hospital’s internet-based resources accessible to support patients and family about pain management. (Yes)	Yes	Yes
		75% panelists agreed that the hospital’s internet-based resources should be continuously available to the nursing team (Yes)	Yes	Yes
	**Quotes**	None	None	
**3**	**3.1 Action statement:** **Develop a practice-oriented, content-specific short training program for pain management**	100% of panelists agreed that a practice-oriented, content-specific short pain management training must be developed. (Yes)	Yes	Yes
	**3.2 Methods**	100% agreed to include specific practice-oriented pain management training content for all nursing care areas (Yes)	Yes	Yes
		Specific methods were: 91.7% agreed that the assessment of patients as important content 83.3% panelists agreed that the selection of appropriate pain strategies based on pain levels should be assessed. (Yes)	Yes	Yes
	**3.3 Responsible persons**	91.7% (n = 11) agreed that on pain nurse specialist should be in every facility (Yes)	Yes	Yes
	**3.3 Time frame**	Only 66.7% agreed on 1 month before the due date of the training program (No)	83.3% consensus was reached that specific practice-oriented pain management content should be included in the program. (Yes)	Yes
	**Quotes**	None	None	Yes
**4**	**4.1 Action statement: Develop a short pain management program that accommodates all learning types.**	100% (N = 12) of the panelists were in agreement. (Yes)	Yes	Yes
	**4.2 Methods**	100% (N = 12) consensus was reached that different learners should be incorporated during training sessions (Yes)	Yes	Yes
		Specific different learner types of nurses were: 100%(n = 12) agreed on creative learners 91.7% agreed to include creative learners 75% (n = 9) agreement on organized thinkers	Yes	Yes
		Specific different learning types to incorporate during pain management training were: 100% (N = 12) agreed to be creative ideas in groups 66.7% (n = 8) of panelists agreed that it should be listening to the information actively, solving different real problems		
	**4.3 Responsible persons**	75% (n = 9) of the panelists agreed that one pain nurse specialist should be responsible (Yes)	Yes	
	**4.4 Time frame**	No consensus. All panelists had diverse views	83.3% agreed that 3 months before the due date of the training program, to ensure learning types are incorporated within the training program (Yes)	Yes
	**Quotes**	None	None	None
**5**	**5.1 Incorporate different teaching approaches to accommodate diverse learners and facilitators in the training of pain management.**	100% (N = 12) panelists agreed that various teaching approaches should be included during the training program. (Yes)	Yes	Yes
	**5.2 Methods**	100% (N = 12) of panelists agreed on the inclusion of different teaching strategies during training (Yes)	Yes	Yes
		Specific teaching strategies to be utilised include: 75% selected engagement in focus groups 66.7% agreed on the use of role-play activities (Yes)	Yes	Yes
	**5.3 Responsible persons**	75% (N-12) consensus selected one pain nurse specialist in every facility. (Yes)	Yes	Yes
	**5.4 Time frame**	No consensus reached, panelists had diverse opinions (No)	75% consensus reached that different teaching strategies to be included to be part of the training program	
	**Quotes**	None	None	None
**6**	**6.1 Action statement: Develop strategies to motivate nurses to participate in the short training program**	100% of panelists agreed that strategies to motivate nurses to participate in the short training program should be developed. (Yes)	Yes	Yes
	**6.2 Methods**	83.3% consensus to involve nurses in the development of learning goals and outcomes for pain management 75% agreed on the involvement of nurses in developing the content of the training (Yes)	(Yes)	Yes
	**6.3 Responsible persons**	83.3% (n = 10) agreed that one pain nurse specialist should be appointed. 75% (n = 9) believed the in all areas of nursing care areas (No)	75% agreed that one pain nurse specialist in every facility to develop the mentioned strategies, motivating nurses’ participation (Yes)	Yes
	**6.4 Time frame**	75% (n = 9) panelists agreed that 1 month before the training program starts was appropriate (Yes)	No consensus on time frame was reached as all panelists had different views	
	**Quotes**	None	None	None
**7**	**7.1 Action Statement: Motivate nurses to apply the knowledge gained in the training program into practice**	100% (N = 12) consensus was reached that nurses must be motivated to apply their knowledge gained in the training program into practice (Yes)	Yes	Yes
	**7.2 Methods**	Several methods were agreed upon to motivate nurses: 100% (N = 12) of nurses should be offered the opportunity to take on the role of pain management. 75% (n = 9) agreed on supporting nurses’ SMART goals and pain management learning 75% (n = 9) agreed that aspects that drive individual nurses to apply what they have learned should be supported (Yes)	Yes	Yes
	**7.3 Responsible persons**	91.7% (n = 11) panelists chose nurse managers in all nursing care areas, 75% (n = 9) of the panelists agreed that clinical directors of nursing operations to be responsible (No)	75% agreed that clinical directors of nursing operations should facilitate the implementation of the methods to motivate nurses to apply their knowledge in practice. (Yes)	Yes
	**7.4 Time frame**	No consensus reached as most panelists had diverse views. (No)	Panelists did not reach an agreement (No)	90% consensus was reached on 1 to 3 months after the training program as a time frame
	**Quotes**	None	None	None

**Table 3 nursrep-15-00442-t003:** Demographic characteristics of the panelists.

Characteristics	Registered Nurses (RNs)		Clinical Facilitators (CFs)		Cumulative	
	**(n=)**	**%=**	**(n=)**	**%**	**(N=)**	**%=**
Gender						
Female	10	83.3	2	16.7	12	100
Males	0	0	0	0	0	0
Education						
Master’s degree	2	16.7	0	0	2	18.7
Bachelor’s degree	5	58.3	2	16.7	9	75
Diploma in Nursing	3	25	0	0	12	100
Nationality						
Filipino	5	41.7	1	8.3	6	50
South African	2	16.7	1	8.3	9	75
Malaysian	2	16.7	0	0	11	91.6
Saudi	1	8.3	0	0	12	100

**Table 4 nursrep-15-00442-t004:** Validated action plan to enhance the transfer of learning of pain management competencies of nurses.

	Action statements						
	**1**	**2**	**3**	**4**	**5**	**6**	**7**
Action Statements	Motivate Nurses to Further Their Studies.	Provide the Nursing Team with Appropriate Pain Management Tools	Develop a Practice-Oriented, Content-Specific Pain Management Short Training	Develop a Short Pain Management Program That Accommodates All Learning Types.	Incorporate Different Teaching Approaches to Accommodate Diverse Learners and Facilitators in the Training of Pain Management.	Develop Strategies to Motivate Nurses to Participate in the Short Training Program.	Motivate Nurses to Apply the Knowledge Gained in the Training Program into Practice.
Methods suggested to achieve the action goal and expected outcomes	Develop a policy to motivate nurses.	Integrate all pain assessment instruments in an electronic patient record system.	Provide pain management training content that focuses on practical application.	Develop a training program that is flexible enough to accommodate a variety of learning types and learning styles used by nurse trainees.	Ensure the training program includes a variety of teaching methods, such as focus groups and role-play.	Include practice-oriented pain management content	Accommodate different learning types and learning styles.
	Propose and advocate for the adoption of the policy to (MNGHA)	Engage nurse supervisors in pain management training.					
	Incorporate the policy into the policies of all hospitals	Ensure that all internet-based resources of hospitals are easily accessible					
		Provide the nursing team with access to all hospitals’ internet-based resources					
Responsible person to implement	Representatives of the nursing policy committee Directors of nursing operations	Clinical Directors	Pain nurse specialist	Pain nurse specialist	Pain nurse specialist	Pain nurse specialist	Nurse managers
Time frame	Four to six months after the MNGHA approves the action plan	To be incorporated into the electronic patient record system within 2–3 months	Three months before the training program’s deadline	Three months before the training program’s deadline	Three months before the date on which the training program is to be executed	One month before the training program’s launch	Within one to three months following the training program

## Data Availability

The primary source of this article is from a completed PhD thesis available at https://uir.unisa.ac.za/handle/10500/31492 (accessed on 5 May 2025). The datasets produced and examined in this study are not publicly accessible to preserve informed consent and confidentiality; however, they can be acquired from the corresponding author upon a reasonable request.

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
