# Peer review of "An Action Plan to Facilitate the Transfer of Pain Management Competencies Among Nurses"

_nursrep, 2025, doi:10.3390/nursrep15120442_

Round 1

Reviewer 1 Report (Previous Reviewer 1)

Comments and Suggestions for Authors

You have satisfactorily addressed all the issues I raised in Review 1.

Author Response

Indicated to have approved all revisions and satisfied

Reviewer 2 Report (Previous Reviewer 2)

Comments and Suggestions for Authors

The manuscript addresses an important and relevant theme: the transfer of pain management competencies among nurses, with a validated action plan derived from mixed-methods evidence and a rigorous Delphi process. The paper is well-structured, comprehensive, and supported by relevant literature.
However, there are some revisions to do to improve the readability of this manuscript:

The background is too extensive and at times repetitive (e.g., pain management not included in curricula is mentioned twice) and the rationale for the study should be sharpened, highlighting what this manuscript adds beyond the author’s PhD thesis.

The link between the mixed-methods approach and the current qualitative focus is not fully clear. The manuscript refers to “previous phases” (Kolobe, 2024 thesis) but does not provide enough methodological detail here. Readers without access to the thesis may struggle to follow. Sampling strategies for nurses and facilitators should be more clearly justified; selection bias should be addressed.

The Delphi process is described, but you should follow CREDES (Conducting and Reporting of Delphi Studies) recommendations.

Quotations from participants are scarce; more qualitative evidence would strengthen trustworthiness. 

While rich in references, the discussion is overly descriptive. A deeper critical analysis of how the validated action plan advances practice compared to existing interventions in other contexts is needed.

Author Response

Comment 1:Changes are made and highlighted in red to improve the background between lines 35 and 200 

Comment 2: We decided to keep the results as they are, as these are based on the final action plan as indicated in Tables 3 and 4

Comment 3: We decided to retain the conclusion  in the same assertions

Comment 4: A repeated statement has been removed  regarding pain management not included in the nursing curricula, lines 101 to 103

Comment 5: This highlights the rationale for conducting the study, which aimed to develop an action plan to address the critical need for structured, practical strategies that enhance nurses' ability to apply pain management knowledge across diverse interdisciplinary settings, Lines 61 to 64.

Comment 6: Figure 3 illustrates the link between the mixed methods. Line 235 to 259.

Comment 7: Rebuttal: We opted for an alternative approach to convey the core message and focus specifically on the developed action plan and validation thereof. Given the extensive nature of the original PhD study, which consisted of five phases and nearly 600 pages, it is not feasible to provide a comprehensive report within the constraints of a journal's page limits. We would also like to emphasize that including excessive amounts of data can make our manuscript less engaging and may impact its appeal to readers. Our intention was to maintain a clear and concise focus on the action plan, a significant contribution aimed at improving pain management competencies, which are essential for quality nursing care. We sincerely hope that the revised manuscript remains focused, accessible, and easy to read, without compromising its quality or diminishing its relevance and value to the journal's readership. 

Comment 8: Sampling technique is clearly detailed in Table 1, lines 284 to 297

Comments 9: Aligned the report with CREDES, lines 127 to 398.

Comment 10: Quotations indicated in the manuscript are those mentioned  only by the participants

Comment 11: Critical analysis improved as added in lines  475 to 477; 502-505; 502-505; 515- 577; 582-477 

Thank you for your feedback and guiding support

Round 2

Reviewer 2 Report (Previous Reviewer 2)

Comments and Suggestions for Authors

The authors have carefully addressed most of the issues raised in the previous review. The methodological description is now clearer, the results section has been better organized, and the discussion has been expanded and supported by updated literature. However, some sentences remain redundant (especially in the Introduction and Discussion).you should resized for better legibility Figures 2 and 3. Please, ensure that the terms “pain nurse specialist” and “clinical facilitator” are used consistently throughout the manuscript. And please verify uniform formatting (especially items [41]–[46]) and ensure all DOIs are active.

Author Response

Thank you for taking the time to review this manuscript once again. We have addressed all previous recommendations; however, one of the reviewer's comments, as noted below, could not be implemented. Please find our responses  in the below along with the corresponding changes highlited in red text in the revised submission.

Comment 1:Results can be improved.

Results kept the same way for maintaining focus.

Comment 2:However, some sentences remain redundant (especially in the Introduction and discussion)

Redundant sentences in the "introduction and discussions" were deleted /rephrased.

Comment 3: You should resize for better legibility, Figures 2 and 3

Figures 2 and 3 resized

Comment 4:Please ensure that the terms "pain specialist and clinical facilitator are used consistently in the manuscript.

Pain nurse specialist and clinical facilitator were consistently used throughout  the manuscript and highlighted in red

Comment 5: And please verify uniform formatting (especially items[ 41]-[46]

Items 41 to 46 are corrected and ensured all DOIs are active, and highlighted all corrections

This manuscript is a resubmission of an earlier submission. The following is a list of the peer review reports and author responses from that submission.

Round 1

Reviewer 1 Report

Comments and Suggestions for Authors

Dear authors,

Thank you for the opportunity to review your manuscript. This study presents a valuable and timely initiative aiming to improve pain management competencies among nurses in Saudi Arabian teaching hospitals. The development and validation of an action plan using a mixed-methods design, framed by the Systemic Model of Transfer of Learning, is a thoughtful and methodologically relevant approach. However, several sections of the manuscript would benefit from more precision, critical depth, and methodological transparency. Below are detailed comments and suggestions, section by section.

abstract
The abstract provides a general overview of the study, but it lacks conciseness and clarity. The phases of the research are described in overly technical terms without clearly linking them to specific outcomes. Consider simplifying the language, especially around the methodology (e.g., explain the sequential explanatory design in more accessible terms) and include clearer information about the structure and content of the final action plan. Quantitative and qualitative findings should be more succinctly stated, with results and implications delineated separately.

introduction
The introduction establishes the relevance of pain management and the need for competency transfer. However, the literature review presented is too descriptive and lacks critical synthesis. There is some repetition regarding knowledge deficits and educational inconsistencies. The rationale for using the Systemic Model of Transfer of Learning could be made more explicit. The operational definition of “competency transfer” should also be more clearly articulated, and the justification for focusing on Saudi Arabian teaching hospitals needs to go beyond accessibility and address the broader applicability of findings.

methods
The study employs a five-phase sequential explanatory mixed-method design, but each phase's methodological details are not consistently presented with sufficient rigor.

  • The sampling approach, although stated as proportionate stratified random sampling, lacks details about how strata were defined and selected.

  • More clarity is needed regarding the instruments used: were they validated tools, or developed by the authors? If the latter, how was their validity and reliability ensured prior to deployment?

  • The Delphi process in Phase 5 is appropriate for expert consensus, but the criteria for panel selection and the number of rounds need clearer justification. Was saturation reached, and how was agreement defined and measured?

  • The action plan table is informative, but the development process (in Phase 4) lacks clarity: how exactly were the data and literature synthesized into action statements?

The manuscript would benefit from the inclusion of a flowchart showing the mixed-method sequence and participant flow across phases.

results
The results are mostly descriptive. The demographic breakdown is adequate, but not meaningfully discussed. For a mixed-methods study, there is insufficient integration of quantitative and qualitative results. For example, panelist feedback in the Delphi validation is reported only in summary (e.g., “consensus reached”), without rich illustrative quotes or deeper analysis of divergent views.

Further, the findings from the earlier phases (1 to 3) are not presented with sufficient detail. Key statistics, such as percentages of nurses who lack access to tools, specific deficits in knowledge areas, or differences in learning styles, are missing. This weakens the evidentiary basis for the developed action plan.

discussion
The discussion generally aligns with the proposed actions but tends to be too confirmatory and references literature without offering critical engagement. Several issues need attention:

  • The discussion of motivation strategies lacks differentiation between intrinsic and extrinsic factors. Were nurses more influenced by professional development, financial incentives, or institutional culture?

  • The role of nurse educators and supervisors is emphasized, but the study lacks data on their perceptions or institutional readiness for implementing such a plan.

  • The limitations section is brief and does not adequately reflect methodological constraints, such as the use of self-administered surveys, reliance on self-report data, or potential response bias in Delphi rounds.

There is limited reflection on broader systemic or cultural barriers to implementation, especially in hierarchical healthcare systems.

conclusion
The conclusions reiterate the study objectives rather than critically evaluating the implications of the action plan. Consider highlighting which parts of the action plan are most feasible or impactful, and which require institutional buy-in. The potential for broader application outside Saudi Arabia is mentioned but not substantiated.

Author Response

Thank you for the comments related to manuscript. We have attended accordingly the comments. Please refer from the attached cover letter.

Reviewer 2 Report

Comments and Suggestions for Authors

This is an important and timely study that addresses a gap in the practical transfer of pain management competencies among nurses in Saudi teaching hospitals. The use of the Systemic Model of Transfer of Learning is appropriate and innovative, and the involvement of stakeholders across all phases is a key strength.

However, to improve the manuscript:

  1. Improve formatting of Table 2: Consider simplifying or splitting complex cells to make it more digestible for readers.
  2. Clarify the Delphi process: Expand slightly on how disagreements or feedback from rounds were handled or modified in the subsequent iteration.
  3. Improve clarity and conciseness of writing: Several sections would benefit from editorial refinement, particularly to reduce redundancy and clarify dense sentences.
  4. Figures and visuals: Improve labeling and resolution of figures. Ensure all elements in Figure 1 are legible.
  5. Policy implications: Consider briefly reflecting on how the findings could be adopted in non-Saudi settings to emphasize generalizability.

Author Response

Thank you for reviewing this manuscript. We have attended the areas to be improved accordingly in red. Please refer from the cover letter attached
